# SCALE EFFICIENTLY: INSIGHTS FROM PRE-TRAINING AND FINE-TUNING TRANSFORMERS

**Yi Tay**[*], **Mostafa Dehghani**[*], **Jinfeng Rao, William Fedus, Samira Abnar, Hyung Won Chung, Sharan Narang, Dani Yogatama**[†], **Ashish Vaswani, Donald Metzler**
Google Research & DeepMind[†]
{yitay,dehghani}@google.com

## ABSTRACT

There remain many open questions pertaining to the scaling behaviour of Transformer architectures. These scaling decisions and findings can be critical, as training runs often come with an associated computational cost which have both financial and/or environmental impact. The goal of this paper is to present scaling insights from pretraining and finetuning Transformers. While Kaplan et al. (2020) presents a comprehensive study of the scaling behaviour of Transformer language models, the scope is only on the upstream (pretraining) loss. Therefore, it is still unclear if these set of findings transfer to downstream task within the context of the pretrain-finetune paradigm. The key findings of this paper are as follows: (1) we show that aside from only the model size, model shape matters for downstream fine-tuning, (2) scaling protocols operate differently at different compute regions, (3) widely adopted T5-base and T5-large sizes are Pareto-inefficient. To this end, we present improved scaling protocols whereby our redesigned models achieve similar downstream fine-tuning quality while having 50% fewer parameters and training 40% faster compared to the widely adopted T5-base model. We publicly release over 100 pretrained checkpoints of different T5 configurations to facilitate future research and analysis.

## 1 INTRODUCTION

Training Transformers incurs both financial and environmental costs (Schwartz et al., 2019; Patterson et al., 2021). To this end, researchers and practitioners often have to work around fixed compute budgets and figure out the best ways to train their models. In lieu of the rising computation demand for training state-of-the-art Transformer (Vaswani et al., 2017; Devlin et al., 2018; Raffel et al., 2019; Brown et al., 2020; Fedus et al., 2021) models, the goal of this paper is to present insights and lessons from scaling Transformers and making them efficient and effective for transfer learning on downstream tasks.

Despite the insights offered in scaling laws research (Kaplan et al., 2020; Hernandez et al., 2021) there remain unresolved questions: Should one follow fixed scaling ratios? If not, should one scale by depth? Or by width? Will scaling experiments on upstream pre-training generalize for downstream transfer? Do scaling protocols for small models generalize to larger models? Are scaling behaviours similar in all compute regions? We hope the insights presented in this paper can be useful to both practitioners and researchers in informing their scaling decisions.

Neural scaling laws (Kaplan et al., 2020) is a common resource that many look to for advice on scaling Transformer architectures. However, this paper limited its scope to an exhaustive study of *upstream* cross entropy on language modeling tasks. It is furthermore unclear if findings from (Kaplan et al., 2020) will transfer to downstream applications. Specifically, Kaplan et al. (2020) proposed that the performance of a Transformer language model strongly depends on model *size* and only weakly on its *shape*. They also argue that many model configurations with the same number of parameters perform similarly regardless of architectural details. Our work empirically confirms this on upstream training but finds a distinct discrepancy when considering practical downstream performance – a key insight that we believe is highly important.

---

[*]Equal contribution

To this end, we conduct extensive experiments involving pre-training and fine-tuning over 200 transformer configurations ranging from 5M to 30B parameters. To the best of our knowledge, this is the largest empirical study of practical scaling of transformer to date that considers *both* upstream and practical downstream transfer. While there have been many proposed scaling protocols for ConvNets (Tan and Le, 2019; Bello et al., 2021), there is still limited advice on scaling of transformer architectures, apart from (Kaplan et al., 2020; Li et al., 2020). Hence, the key goal of this paper is to distill our experiences and insights with scaling Transformer architectures and share them with the broader community.

**Contributions** The overall findings and insights of the paper can be summarized as follows:

- We find that scaling laws may differ in upstream and downstream setups. Specifically, contrary to Kaplan et al. (2020), we find that downstream performance *strongly* depends on shape and not only on model size. Hence, pretraining performance may not necessarily transfer to downstream applications. (Figure 1).

- Our findings show that pre-training perplexity can often be a deceiving indicator of downstream quality and therefore model building based on upstream perplexity can be challenging. Scaling laws can differ substantially when considering metrics on actual downstream fine-tuning. (Figure 1)

- Given that empirical scaling laws differ when considering quality on the downstream, our work investigates the pareto-frontier of transformer configurations in this setup. We find that the canonical model configurations such as *T5-Base* and *T5-Large* sizes (Raffel et al., 2019) are relatively inefficient (Figure 2). Note that these sizes are based off the canonical BERT (Devlin et al., 2018) base and large sizes.

- We find that scaling strategies differ at different compute regions, i.e., applying same strategies at different compute regions (small vs large) has a different effect on model quality. This has practical implications since finding strategies at small scale might not necessarily transfer or generalize to higher compute regions (section 4.2).

- After extensive empirical exploration of the pareto-frontier of transformer models, we propose a simple but effective scaling strategy which we call the *DeepNarrow strategy*. We show that we are able to obtain model quality on par or better than canonical model sizes (e.g., base) with $50\%$ less parameters and being $40\%$ faster. While we highlight the limitations of this strategy, we also show that this *DeepNarrow* strategy is applicable to all model sizes. (Table 4).

- To consider how generalized these scaling strategies are, we conduct additional experiments on Vision Transformers (ViT; Dosovitskiy et al., 2020) to verify them in the vision domain. Moreover, on top of the 17 GLUE (Wang et al., 2018) / SuperGLUE (Wang et al., 2019) and SQuAD (Rajpurkar et al., 2016) tasks we employed in our extensive study, we verify our findings via additional downstream experiments across 12 diverse language tasks (section .2).

- We release (1) the pre-trained checkpoints for our T5 models with improved scaling protocols and (2) all 100+ model checkpoints, including intermediate training checkpoints to the research community. We believe that this is a treasure trove of data to study the behaviour of large LM pretraining and finetuning especially pertaining to scaling laws. The checkpoints and code will be released at `https://github.com/google-research/google-research/tree/master/scaling_transformers`. The checkpoints are now publicly available at our Google Cloud Bucket `gs://scenic-bucket/scaling_explorer/scaling_explorer`. More recently, these checkpoints are also now available on Huggingface `https://huggingface.co/models?other=deep-narrow`.

## 2 RELATED WORK

Transformers (Vaswani et al., 2017) have become ubiquitous in the modern deep learning stack and have seen widespread impact across not only language (Devlin et al., 2018; Raffel et al., 2019; Brown et al., 2020) but also computer vision (Dosovitskiy et al., 2020; Arnab et al., 2021), reinforcement

Table 1: Table of model configurations. $N_L$ is the number of layers, $d_{ff}$ is the size of the MLP, $d_{model}$ is the hidden size of the model. $d_{kv}$ is the size of each key-value vector. $N_H$ is the number of heads. $P$ is the default model parallelism.

| Model | $N_L$ | $d_{ff}$ | $d_{model}$ | $d_{kv}$ | $N_H$ | #Params |
|---|---|---|---|---|---|---|
| Tiny | 4/4 | 1024 | 256 | 32 | 4 | 16M |
| Mini | 4/4 | 1536 | 384 | 32 | 8 | 31M |
| Small | 6/6 | 2048 | 512 | 32 | 8 | 60M |
| Base | 12/12 | 3072 | 768 | 64 | 12 | 220M |
| Large | 24/24 | 4096 | 1024 | 64 | 16 | 738M |
| XL | 24/24 | 16384 | 1024 | 128 | 32 | 3B |
| XXL | 24/24 | 65536 | 1024 | 128 | 128 | 11B |
| XXXL | 28/28 | 131072 | 1280 | 128 | 256 | 30B |

Table 2: Description of different knobs used in the paper to define scaling operations.

| Scaling Op | Description |
|---|---|
| NL | Num. layers |
| EL | Num enc. layers |
| DL | Num. dec. layers |
| DM | $d_{model}$ |
| KV | $d_{KV}$ |
| NH | Num. of heads |
| FF | $d_{ff}$ |
| SH | Shared heads |
| SKV | Tied key-values |

learning (Parisotto et al., 2020) and computational biology (Senior et al., 2020). To this end, discovering empirical scaling laws of these models is a research area that has garnered considerable interest (Kaplan et al., 2020; Henighan et al., 2020; Hernandez et al., 2021; Bahri et al., 2021).

Discovering empirical scaling laws that govern neural language model scaling has been a recent subject of keen interest (Kaplan et al., 2020; Hernandez et al., 2021; Bahri et al., 2021). Many of these works present scaling laws across a variety of axis such as model size, compute and/or dataset size. It is worth to note that many of these works evaluate on autoregressive language modeling and use cross entropy loss to measure performance (Kaplan et al., 2020; Hernandez et al., 2021). There are a multitude of interesting findings presented (Kaplan et al., 2020) whereby the authors show that performance (loss) scales as a power-law with model size and dataset size. However, one notable claim is that architectural details (e.g., network depth and width) have minimal effects. Subsequently, Hernandez et al. (2021) builds upon the work of Kaplan et al. (2020), evaluating scaling laws for 'transfer'. To this end, the authors study the effect of dataset scaling on unsupervised transfer learning and finetuning. That said, the experiments of Hernandez et al. (2021) are mainly targeted at dataset transfer between two different distributions (language and code) and make the same assumptions as Kaplan et al. (2020) about model scaling. In a similar vein, Henighan et al. (2020) studied empirical scaling laws for different domains for generative modeling in vision, video and multimodal setups.

There have been increasing demand for training and scaling Transformers (Shoeybi et al., 2019; Raffel et al., 2019; Fedus et al., 2021; Conneau et al., 2019; Lin et al., 2021). Despite the benefits from improved performance, there are financial considerations and environmental costs (Schwartz et al., 2019; Patterson et al., 2021) to training these models. Given that every moment spent on hardware accelerators is a cost incurring activity, we believe that research in distilling practical scaling insights and recommendations to be highly crucial (Li et al., 2020; Kaplan et al., 2020; Bello et al., 2021).

Notably, the research problem of making transformers efficient have also been tackled from an extensive number of angles such as (but not limited to) distillation (Hinton et al., 2015), compression (Zafrir et al., 2019), parameter sharing (Lan et al., 2019; Tay et al., 2019; Zhang et al., 2021), efficient attention (Tay et al., 2020c; Kitaev et al., 2020; Choromanski et al., 2020; Tay et al., 2020b; Ainslie et al., 2020; Jaegle et al., 2021), architecture search (So et al., 2019), alternative non Transformer-based architectures (Tolstikhin et al., 2021; Tay et al., 2021a; 2020a; Lee-Thorp et al., 2021). With so much extensive research into novel techniques to improving the efficiency of transformers, it is surprising that the standard configurations (e.g., base, large) of transformers in BERT (Devlin et al., 2018) or T5 (Raffel et al., 2019) have not been rethought.

## 3 METHODS

This section describes our main experimental setup.

**Architecture** We study a Transformer encoder-decoder architecture that uses relative attention based of the T5 architecture (Raffel et al., 2019). The choice of adopting Seq2Seq architec-

tures (Sutskever et al., 2014) is mainly due to their universality and ability to both subsume encoder (Bert-like) and decoder (language) models within an identical framework. Moreover, the universality of Seq2Seq architectures also allow us to fine-tune across a broad range of tasks. Our implementation and experiments are performed in Mesh Tensorflow[1] (Shazeer et al., 2018) using the default T5 library[2].

**Model Configurations**   We first define eight Transformer sizes, i.e., tiny, mini, small, base, large, XL, XXL and XXXL. The small, base, large, XL and XXL corresponds to the canonical T5 sizes that are released in Raffel et al. (2019). We use three other sizes as starting points, e.g., tiny and mini since there is a lack of representation of transformers at lower compute regions.

**Pretraining**   We pretrain on the Colossal Cleaned Common Crawl Corpus (C4; Raffel et al., 2019). We pre-train encoder-decoder models using the span-based masked language modeling (MLM) objective (Fedus et al., 2018; Devlin et al., 2018). We pretrain all our models for $2^{19}$ steps using 16 TPU-v3 chips. For larger models, we run our models with $64$ TPU-v3 chips. We use $2^{19}$ steps since majority of the experiments in (Raffel et al., 2019) were conducted in the same fashion. We would also like to emphasize that the official released T5 checkpoints were pretrained on $1T$ tokens (1 million steps with a batch size of 2048). Given that this extended long pretraining setup is infeasible given the number of experiments we would have to run, we opt to follow the standard ablation setup in (Raffel et al., 2019) which pretrains on more manageable number of tokens.

**Downstream Tasks**   We consider a myriad of downstream tasks.   In total, we consider 17 tasks. We finetune on a mixture of GLUE (Wang et al., 2018), SuperGLUE (Wang et al., 2019), SQuAD (Rajpurkar et al., 2016) for the key downstream experiment results and report aggregate GLUE/SuperGLUE scores. We believe that an aggregate of 17 tasks in natural language understanding that conmprises of both high-resource and low-resource tasks gives us a good overview of a model's downstream performance. Finetuning is typically performed with 16 TPU-v3 chips.

**Notation for Scaling Operators**   For the remainder of the paper, we use a shortform code for each scaling operator applied on a standard transformer size. For example, NL32-SM refers to scaling small (SM) transformers to 32 layers (NL32). We use EL,DL to represent scaling encoder and decoder layers independently, KV to represent scaling each key-value size, DM to represent scaling $d_{model}$. NH to represent modifying the number of heads and FF to represent scaling $d_{FF}$. The initial/standard model sizes are tiny, mini, small, base, large, XL, XXL and XXXL. This is described in Table 2.

**Convention**   With the exception of Figure 1, all charts are plotted with FLOPS as the main compute metric.   We use number of params for Figure 1 to align with Kaplan et al. (2020).   All of the downstream results are plot with SuperGLUE accuracy (Wang et al., 2019) as the Y-axis. Due to the lack of space, we report charts/plots of other metrics (params of speed) and other tasks (GLUE or SQuAD) in the supplementary material. All parameter counts also include the embedding parameters. We re-emphasize that it is critical to take into account multiple facets of efficiency and therefore report all three key metrics (FLOPs, throughput/speed and parameter count) in the supplementary material.

**Model and Data Parallelism**   By default, our models are trained across multiple workers via data parallelism. As per convention in the T5 paper, our larger models use the default model parallelism. Specifically, this is set to 2 for large models, 8 for XL models and 32 for XXL models.

## 4   ANALYSIS AND RESULTS

This section presents our overall findings and key results.

---

[1]https://github.com/tensorflow/mesh
[2]https://github.com/google-research/text-to-text-transfer-transformer

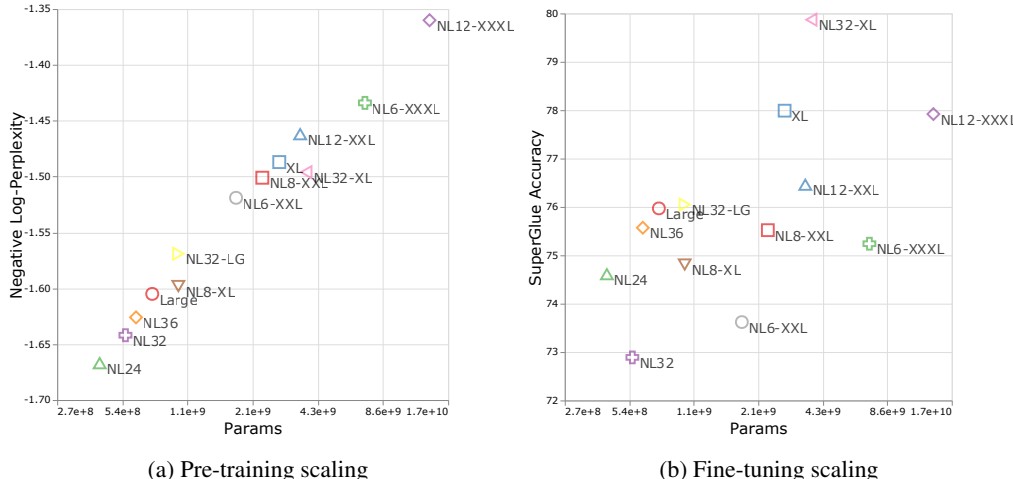

(a) Pre-training scaling            (b) Fine-tuning scaling

Figure 1: **The predictability and unpredictability of pre-training versus fine-tuning.** While the upstream pre-training performance measured by negative log-perplexity scales with model size quite independently from the model shape, the downstream performance (SuperGlue (avg) score) does not. This indicates that the shape of models plays an important role on how it performs on the target task and the performance is not merely a function of parameter size.

## 4.1 MODEL SHAPE MATTERS

We extend the results of Kaplan et al. (2020) to fine-tuning and present model shape dependence not highlighted in Hernandez et al. (2021). Kaplan et al. (2020) studies pre-training (upstream) and concludes that performance depends only weakly on model shape, but strongly on model size. Hernandez et al. (2021) extends this work to measure an effective data transfer measure when pre-training and then fine-tuning on a Python dataset. However, this work does not consider details of model shape, and instead focused on the relative predictability with model scale alone. Our work stands in contrasts since we find that model shape matters considerably for downstream fine-tuned performance.

Figure 1 shows compute-performance scatter plots for pre-training (left) and fine-tuning (right) over a dozen Transformers. The models considered are sampled diversely within a two-order of magnitude band of model scales. We adjust the model shape primarily through depth variations, starting with configurations such as XXXL (33B), XXL (11B), XL (3B) and LG (750M) parameters but have their depths/lengths modified. From Figure 1 reveals a strong correlation of the upstream performance with model size, corroborating the neural scaling laws of Kaplan et al. (2020). But the strong pre-training correlation largely vanishes when fine-tuning these models on SuperGLUE (Wang et al., 2019). While we confirm the findings of Kaplan et al. (2020) that performance scales strongly dependent on model size but weakly on model shape, we find that model shape (such as depth-scaling) is highly important for downstream transfer – a dimension that is not considered in Kaplan et al. (2020).

As a substantiating point and additional context to Figure 1, we also show via a counter-example that pretraining perplexity is not indicative of transfer performance, i.e., we explicitly show that a case (in Table 3) where a model can have outstanding pre-training perplexity but substantially undeliver when it comes to downstream performance. To the best of our knowledge, while this has been mentioned implicitly in several existing works (Narang et al., 2021), this is the first work explicitly shows this point.

**Zooming in Versus Zooming Out**     Here, one may argue that a general trend (even on downstream) may still exist if we zoom out and cover a very wide range of model sizes (e.g., very tiny to very large). A tiny model is not likely to outperform a very large model no matter how well-configured it might be. Our purpose here is not to contradict this general trend but to distinguish between both arguments. We argue that, in practical setups, comparisons between models and scaling decisions are often made when *zooming-in* and our pairwise comparisons above are not on largely different

Table 3: **Upstream performance does not guarantee downstream performance.** Example points from Figure 1. A model with improved upstream quality (as measured by validation perplexity) can do significantly worse on transfer if the shape setting is not right. Hence, pre-training perplexity can be misleading.

| Name | $N_L$ | $d_{ff}$ | $d_{model}$ | $d_{kv}$ | $N_H$ | #Params | PPL | Downstream |
|------|-------|----------|-------------|----------|-------|---------|-----|------------|
| NL12-XXL | 12 | 65536 | 1024 | 128 | 128 | 3.6B | -1.46 | 85.1/76.5/88.1 |
| NL32-XL | 32 | 16384 | 1024 | 128 | 32 | 3.8B | -1.49 | 86.9/79.9/89.5 |

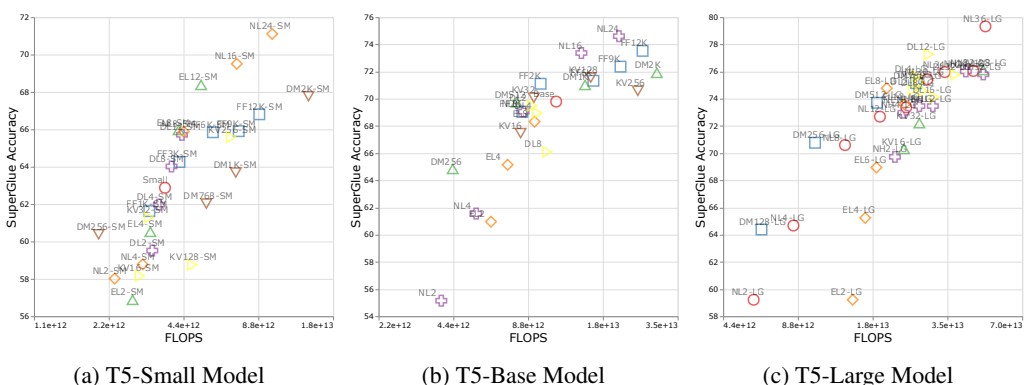

|(a) T5-Small Model|(b) T5-Base Model|(c) T5-Large Model|

Figure 2: **Downstream scaling properties is scale-dependent**. The downstream performance on SuperGLUE has qualitatively different scaling properties across models sizes. From left to right, we fine-tune model configurations closely matched to T5-Small, T5-Base and T5-Large.

models, rather those that are on the same neighborhood in the size (close in the x-axis). Thus, what we claim is that when you zoom in, which is what happen in practice, it is not uncommon to see cases similar to the models in Table 3 where taking the upstream perplexity into account may lead to a sub-optimal choice. It is also worth to mention that zoom-ing in on upstream returns very different trends compared to zoom-ing in on downstream results.

## 4.2    Scaling behaviour at different compute regions is different

In this section, we evaluate how each scaling hyperparameter and model shape influences a model's position on the compute-performance chart. Figure 2 shows three plots which varying different scaling knobs. Given three starting points, small, base and large, we scale the starting points across different knobs. It is clear from Figure 2 that the effect of applying different scaling operators is very different across different compute regions. We also observe that the Pareto-boundary is also very different at different compute regions. The implications of this finding is nontrivial, since this effectively means that finding improvements at a (plausibly) smaller scale in hopes that it will generalize at large scale might not be an effective strategy. This corroborates recent findings of Bello et al. (2021) which studied scaling properties of ImageNet models. Their paper demonstrates that the compound scaling rules (Tan and Le, 2019) derived in a small-scale regime, lead to Pareto-inefficient models when then extrapolated to larger scales.

## 4.3    Not all Scaling Strategies and Model Shapes are Created Equal

From Figure 2, we can also observe that different scaling strategies results in very different outcome. A common pattern amongst all three model sizes is that the $NL$ operator has strong influence on the Pareto-frontier. On the other hand, settings such as KV (varying $d_{kv}$) seem to result in models that are less Pareto-efficient. We notice mixed signals from varying $d_{model}$. In this case, scaling down results in a model on the pareto-frontier, but scaling up, i.e., $DM2K$ results in a highly pareto-inefficient model. When compared to scaling up, $NL$ is a substantially superior option to $d_{model}$. We describe scaling recommendations in subsequent sections. Overall, it does seem like $d_{kv}$ and $N_H$ does not influence the Pareto frontier as much as other knobs.

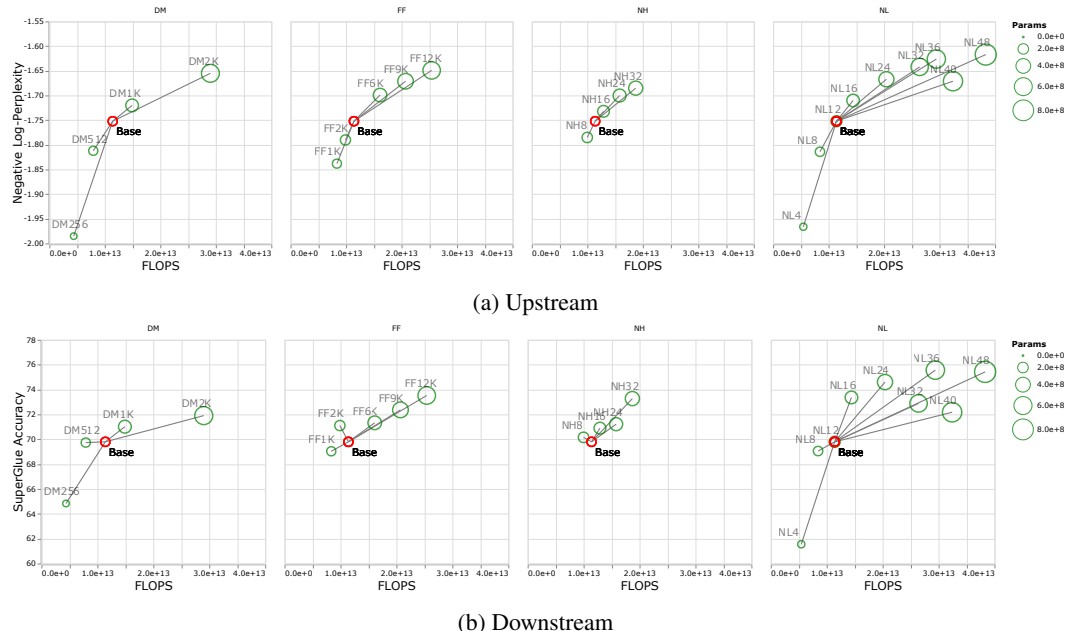

(a) Upstream

(b) Downstream

Figure 3: Different scaling with respects to different knobs, in upstream and downstream. On the plots, **DM** refers to scaling model dimension, **FF** refers to scaling FFN hidden size, **NH** is number of heads, and **NL** is number of layers. Size of each circle indicates the model size in terms of number of trainable parameter parameters.

**Effect of scaling different knobs** Figure 3 illustrates the effect of scaling different knobs on the compute-performance boundary. It becomes clear that not all strategies influence the boundary with the same impact. For example, *NL* has the biggest impact while *NH* does not influence the model's position on the graph much. Finally, we also note that the effect of scaling on upstream and downstream is quite different. For example, FF2K is clearly better option than the canonical base model in downstream but not upstream.

**Scaling Depth vs Width** In Figure 3, we also note that the **NL** scaling operator (depth) has generally impact on the Pareto-boundary as compared to width (**FF**). For instance, we can see that FF12K (in Figure 3 is clearly not Pareto-optimal, being outclassed by configurations such as NL16-SM, EL12-SM. Likewise in the base setup, FF9K and F12K are less Pareto-efficient as compared to NL16 and NL24.

## 4.4 SCALING RECOMMENDATIONS

We generally recommend a *DeepNarrow* strategy where the model's depth is preferentially increased[3] before considering any other forms of uniform scaling across other dimensions. This is largely due to how much depth influences the Pareto-frontier as shown in earlier sections of the paper. Specifically, a tall small (deep and narrow) model is generally more efficient compared to the base model. Likewise, a tall base model might also generally more efficient compared to a large model. We generally find that, regardless of size, even if absolute performance might increase as we continue to stack layers, the relative gain of Pareto-efficiency diminishes as we increase the layers, converging at 32 to 36 layers. Finally, we note that our notion of efficiency here relates to any one compute dimension, i.e., params, FLOPs or throughput (speed). We report all three key[4] efficiency metrics (number of params,

---

[3]Our concurrent work Charformer (Tay et al., 2021b) makes use of a DeepNarrow inspired strategy which is referred to as Tall in the paper.

[4]It is often assumed that number of parameters, speed and FLOPs tend to correlate. We find that this is not always the case especially when dealing with modeling choices that influences parallelism (depth vs width). Therefore, we emphasize the importance of reporting all key efficiency metrics.

Table 4: Efficient *DeepNarrow* alternatives to the canonical T5 model sizes using the *DeepNarrow* strategy. Models are all Pareto efficient at least to one or more aspect of compute and one or more downstream task. XXL and XL32L models are trained on 64 TPU-V3 chips and so they are faster.

| Model | #Params | #TFlops | Steps/s | Ppl (C4) | GLUE | SGLUE | SQuAD | AVG |
|---|---|---|---|---|---|---|---|---|
| Small | 61M | 3.7 | 23 | -2.021 | 77.45 | 62.88 | 80.39 | 73.57 |
| Mini-8L | 50M | 3.2 | 24 | -2.056 | 77.11 | 63.35 | 80.12 | 73.52 |
| Base | 223M | 11 | 9 | -1.752 | 82.53 | 69.80 | 85.14 | 79.16 |
| Small 16L | 134M | 7.2 | 13 | -1.825 | 82.57 | 69.51 | 84.12 | 78.73 |
| Small 20L | 164M | 8.6 | 11 | -1.798 | 83.22 | 69.44 | 85.23 | 79.30 |
| Small 22L | 179M | 9.3 | 10 | -1.798 | 82.52 | 70.68 | 85.39 | 79.54 |
| Small 24L | 193M | 10 | 9 | -1.783 | 83.11 | 71.11 | 85.45 | 79.92 |
| Small 32EL | 143M | 10 | 10 | -1.897 | 82.77 | 70.66 | 86.01 | 79.81 |
| Large | 738M | 34 | 4 | -1.605 | 85.08 | 75.97 | 87.55 | 82.87 |
| Base 36L | 621M | 29 | 3 | -1.626 | 85.26 | 75.57 | 87.84 | 82.89 |
| XL | 2.9B | 64 | 1 | -1.487 | 86.49 | 77.99 | 88.70 | 84.38 |
| Large 36L | 1.1B | 50 | 2 | -1.564 | 87.22 | 79.34 | 89.21 | 85.27 |
| XXL | 11.3B | 367 | 1 | -1.430 | 86.91 | 79.20 | 89.50 | 85.20 |
| XL 32L | 3.8B | 169 | 3 | -1.500 | 86.94 | 79.87 | 89.46 | 85.42 |

FLOPS and speed) and leave this decision to the practitioner to decide which compute dimension to consider.

**Efficient Alternatives to T5-Base/Large/XL/XXL** Table 4 describes this phenomena in which we list efficient alternatives to the canonical model sizes using the *DeepNarrow* strategy. Note that this list is not exhaustive. Firstly, we find that significantly increasing the depth of the small model does substantially better in terms of the compute-performance trade-off and may result in pareto-efficient models. The Small 16L model achieves comparable performance to Base while being $40\%$ faster, cost $50\%$ less parameters and has only about $63.1\%$ of total FLOPs. Alternatively, the Small 24L model has $87\%$ of FLOPs of the base model, similar speed (steps/s), and only $16\%$ parameter savings and yet outperforms Base on all three downstream tasks. Meanwhile the canonical large model can be outperformed by a base model of 36 layers with $16\%$ parameter saving and lower flops cost. The Large 36L model is only $37\%$ of the parameter cost of the XL model and yet outperforms the XL model on all three downstream tasks. Finally, the XL 32L model is only a **third** the size of the XXL model, approximately consume $44\%$ the number of FLOPs of the XXL model and is about 3 times faster on the same hardware.

**The Limits of Depth vs Width** We note an obvious limitation with our advice. Scaling depth has an obvious limiter, i.e., they are non-parallelizable across different machines or devices and every computation has to always wait for the previous layer. This is unlike width, which can be easily parallelizable over thousands or hundreds of thousands of devices. Within the limitation of scaling to 64 workers with a model parallelism of 32, we still find that scaling depth can still improve the Pareto-efficiency of models. From our experiments, from Table 4, we see that the efficient small DeepNarrow models (e.g., Small 16L etc) are still much faster than the base models. Things get tricky as we approach larger models where model parallelism makes it difficult to compare the utility between wide and deep models. To this end, we believe the proposed scaling protocol holds within a certain hardware limit. Scaling to extreme number of machines via model parallelism (of width) is out of scope of this paper. Another potential drawback to depth-scaling is that this may influence the stability of training these models (due to vanishing gradients). However, we did not observe this in our experiments with T5 models.

**Relationship of Model Depth with Pareto-frontier** Figure 2 shows the performance of scaling small, base and large models by depth. It is clear that the small model (green) dominates the Pareto-frontier initially but slowly tapers off at a certain point. Here, we note that the depth-scaled small model is more Pareto-efficient compared to the base model. After the Pareto-efficiency of the small

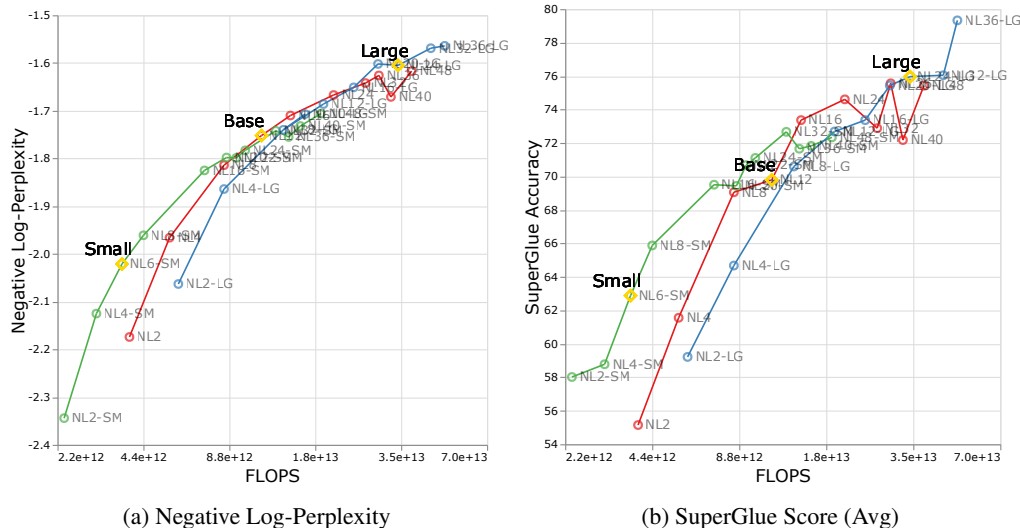

(a) Negative Log-Perplexity

(b) SuperGlue Score (Avg)

Figure 4: Compute-Performance trade-off when scaling model depth of different starting points (Small, Base, and Large).

model tapers off, the base model (red line) becomes Pareto-efficient. Similarly, this tapers off and the large model becomes Pareto-efficient.

## 5 CONCLUSION

In this paper, we present several important findings pertaining to training and practical usage of efficient transformer architectures. Specifically, we discovered that scaling laws differ in upstream and downstream. Contrary to prior work (Kaplan et al., 2020), we emphasize the importance of model shape for ensuring strong downstream performance. Next, we also discovered that scaling happens rather differently at different compute regions and scaling a small model would behave differently from scaling a large model. We highlight that this has implications since model development in a certain compute region could potentially not transfer to another compute region. We go on to show that not all model knobs are created equal, i.e., some knobs (such as depth) has a larger impact on the Pareto-frontier. Finally, we propose a simple but highly effective improved scaling protocol. With this strategy, we obtain models with similar downstream finetuning quality while having 50% fewer parameters and/or training 40% faster.

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

APPENDIX

### .1 TRANSFERABILITY OF RESULTS TO VISION TRANSFORMERS (VIT)

Following our language experiments, and as per the advice of (Narang et al., 2021), in order to examine our *DeepNarrow scaling strategy* in another domain and check if the observations extends to the cases where Transformers are applied on other modalities, we pre-retrained several different configurations of Vision Transformer (ViT; Dosovitskiy et al., 2020) and evaluated on downstream few-shot image recognition task. We focused on investigating the pareto-efficiency of DeepNarrow small models compared to base models.

We follow the exact same setup as Dosovitskiy et al. (2020) and pre-train ViT on the JFT dataset (Sun et al., 2017) with 18k classes and 303M images, for 7 epochs. We evaluate our model on ImageNet 10-shot classification. In our experiments, we use the patch size of $32 \times 32$.

Table 5: Results on image recognition task. All models are trained with the same batch size using 64 TPU-V3 chips.

| Model | #Params | GFLops | Steps/s | ImageNet-10Shot |
|---|---|---|---|---|
| ViT-S | 62M | **1.37** | **8.83** | 45.3 |
| ViT-B | 102M | 4.44 | 6.74 | 58.9 |
| ViT-S$_{L=24}$ | **87M** | 3.94 | 6.11 | **59.7** |
| ViT-S$_{L=28}$ | **99M** | 4.58 | 5.36 | **61.6** |

**Results** Table 5 report results on ViT experiments. When considering the number of trainable parameters or FLOPs, we observe that DeepNarrow scaling of the ViT-S model achieves better Pareto efficiency compared to the ViT-B model. Notably, when $L = 24$, the model achieves better few-shot accuracy with $15\%$ less parameters, $11\%$ less FLOPs and achieves $+1.4\%$ percentage improvement in accuracy. With respect to the step per seconds (speed), given ViT-S$_{L=24}$ or ViT-S$_{L=28}$ add to the sequential operations in depth, they become a bit slower than ViT-B. However, we consider 6.11s and 6.74s to be reasonably within the same ballpark. In short, the ViT-S$_{L=24}$ is still a compelling alternative.

### .2 HOW TRANSFERABLE ARE THESE RESULTS ACROSS MULTIPLE DIVERSE NLP TASKS?

To verify that our scaling protocols transfer to tasks outside of the 17 tasks explored in (GLUE/SuperGLUE and SQuaD), we run additional experiments on a myriad of diverse NLP tasks. Verifying whether the findings generalize outside GLUE and SuperGLUE is important since we do not want to fall prey to the *benchmark lottery problem* (Dehghani et al., 2021). As such, the purpose of this additional experiment is to verify if our results are universal enough for a general recommendation.

**Setup** In this experiment, we compare the base T5 transformer with the efficient 24 layer small model using the *DeepNarrow* strategy, which has $14\%$ less parameters and $10\%$ less FLOPS compared to the T5 base model. This finetuning protocol uses a constant learning rate of $10^{-3}$ and a batch size of 128 for all tasks. Note that we used the same pretrained models as earlier sections that produced finetuned results on SuperGLUE and GLUE.

**Diverse NLP tasks** We conduct experiments on a total of 12 extra tasks, i.e., 6 tasks of Rainbow (Lourie et al., 2021) which contains commonsense reasoning tasks, 3 generation/summarization tasks (XSum (Narayan et al., 2018), CNN/Dailymail (See et al., 2017) and MultiNews (Fabbri et al., 2019)), along with 3 text classification tasks (civil comments (Borkan et al., 2019), wiki toxicity (Wulczyn et al., 2017) and IMDb reviews (Maas et al., 2011)). For the Rainbow suite, we co-train all tasks in (Lourie et al., 2021) and report peak validation results.

Table 6: Rainbow dataset.

| Task | Base | DeepNarrow |
|------|------|------------|
| ANLI | **65.7** | **65.7** |
| CosmoQA | 69.9 | **70.0** |
| HellaSwag | **49.7** | 48.9 |
| PQA | 73.7 | **74.4** |
| SQA | 65.1 | **66.0** |
| Wino | 65.3 | **65.9** |

Table 7: Generation tasks (Rouge-L).

| Task | Base | DeepNarrow |
|------|------|------------|
| XSum | 32.3 | **33.1** |
| CNN/Dailymail | **38.9** | **38.9** |
| MultiNews | 20.2 | **20.5** |

Table 8: Classification tasks (Acc).

| Task | Base | DeepNarrow |
|------|------|------------|
| CivilComments | 88.2 | **88.4** |
| WikiComments | **95.4** | **95.4** |
| IMDb | **94.4** | **94.4** |

**Results on other tasks** Table 6, Table 7 and Table 8 reports results on these 12 tasks. On all 12 additional diverse NLP tasks considered , we show that the Pareto-efficient alternative outperforms or ties with the base T5 model on 11 out of 12 tasks where 4 of them are ties. It is good to bear in mind that this is a model which is effectively smaller in parameters and has 10% less FLOPs.

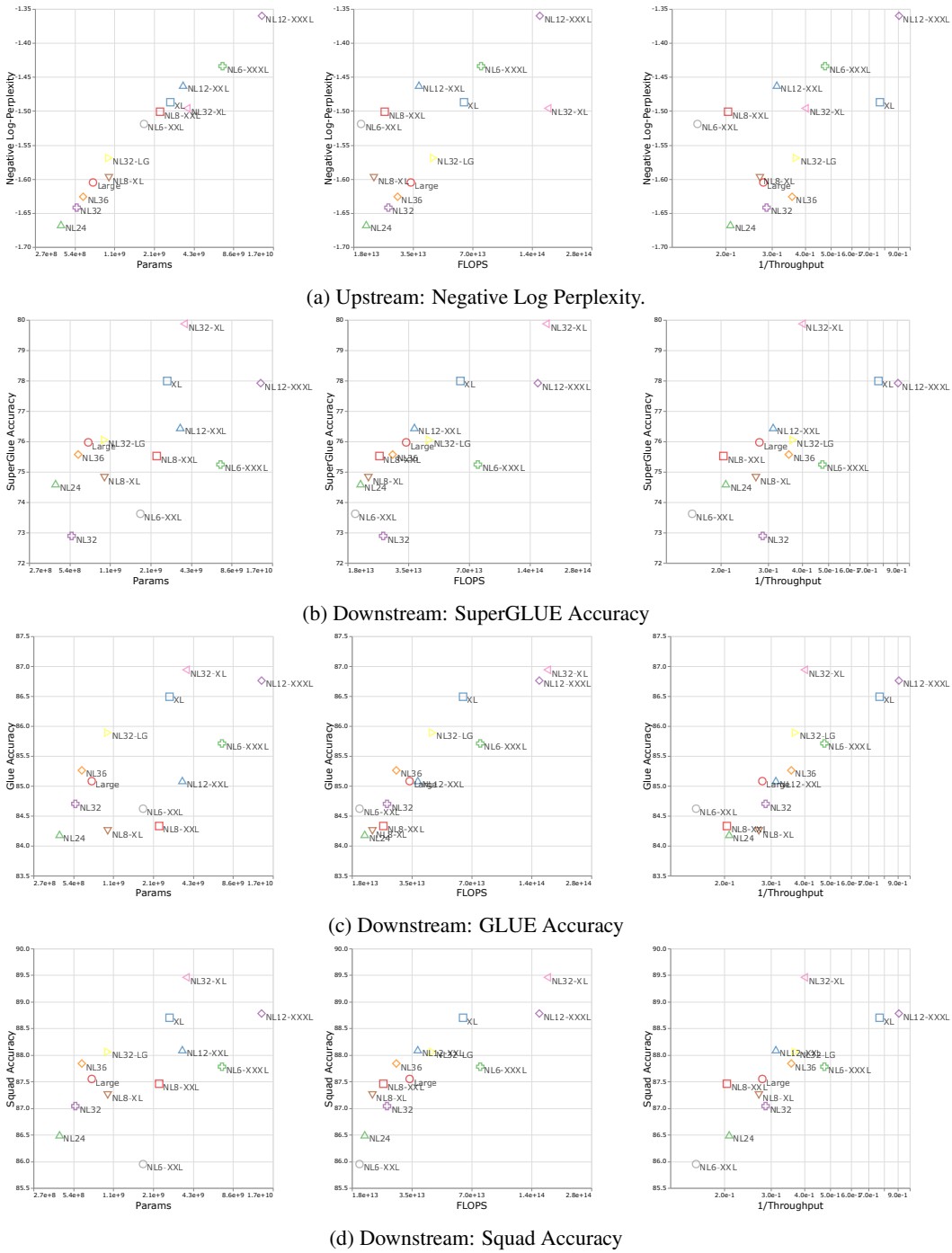

(a) Upstream: Negative Log Perplexity.

(b) Downstream: SuperGLUE Accuracy

(c) Downstream: GLUE Accuracy

(d) Downstream: Squad Accuracy

Figure 5: Performance on upstream and different downstream tasks with respect to number of parameters, FLOPs, and throughput for models presented in Figure 1b.

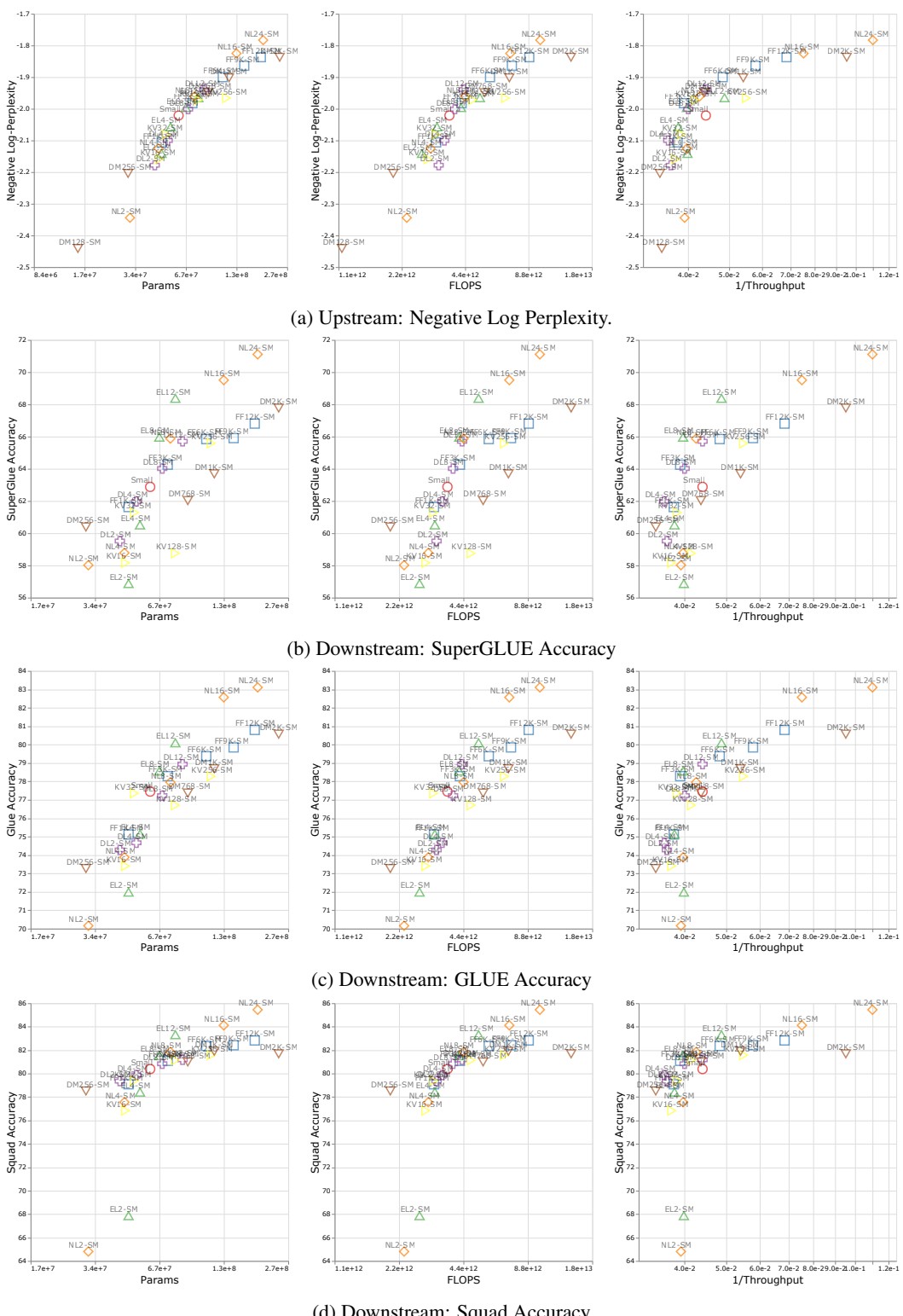

(a) Upstream: Negative Log Perplexity.

(b) Downstream: SuperGLUE Accuracy

(c) Downstream: GLUE Accuracy

(d) Downstream: Squad Accuracy

Figure 6: Performance on upstream and different downstream tasks with respect to number of parameters, FLOPs, and throughput for small models presented in Figure 2a.

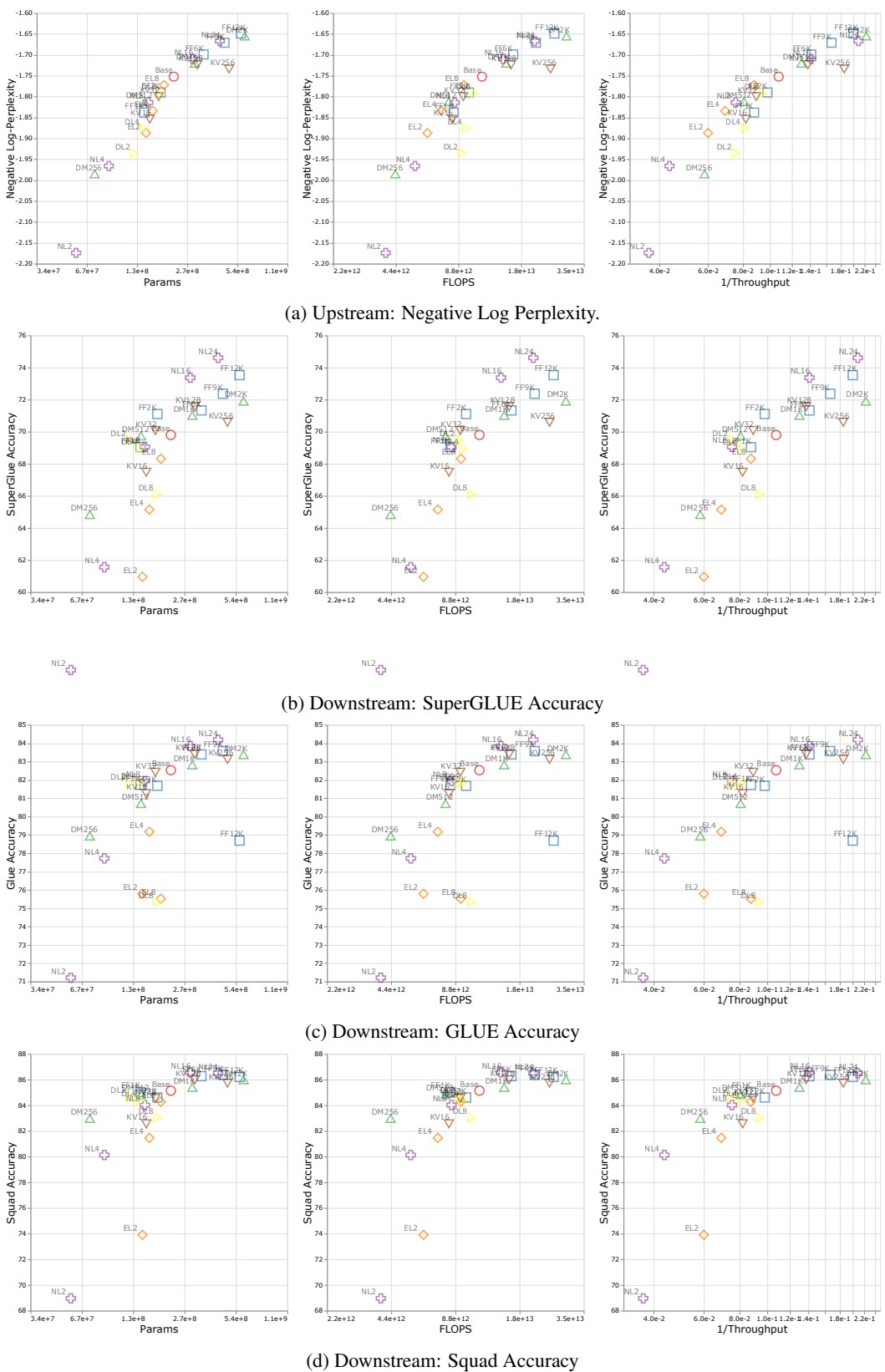

(a) Upstream: Negative Log Perplexity.

(b) Downstream: SuperGLUE Accuracy

(c) Downstream: GLUE Accuracy

(d) Downstream: Squad Accuracy

Figure 7: Performance on upstream and different downstream tasks with respect to number of parameters, FLOPs, and throughput for base models presented in Figure 2b.

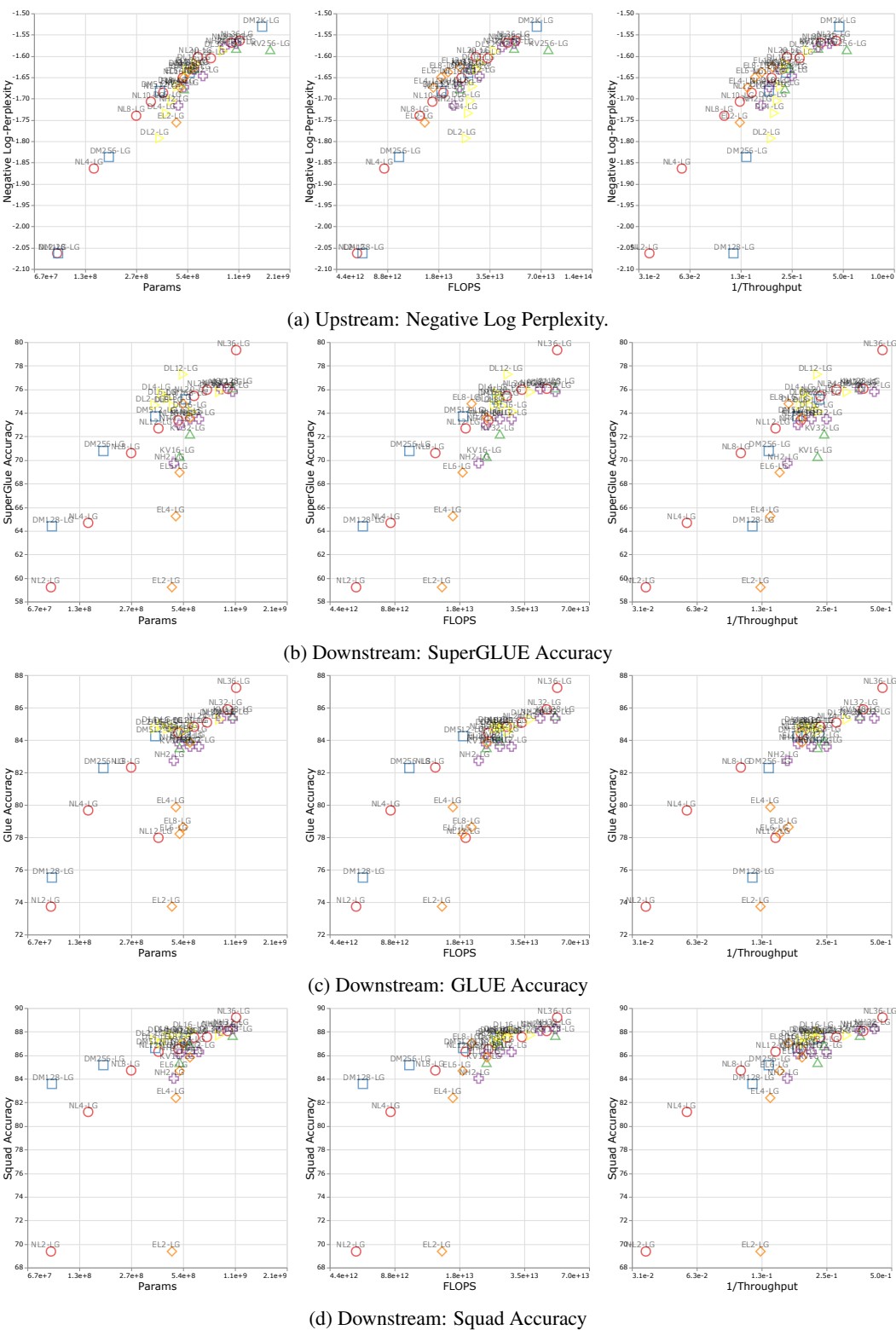

(a) Upstream: Negative Log Perplexity.

(b) Downstream: SuperGLUE Accuracy

(c) Downstream: GLUE Accuracy

(d) Downstream: Squad Accuracy

Figure 8: Performance on upstream and different downstream tasks with respect to number of parameters, FLOPs, and throughput for large models presented in Figure 2c.

