# OpenReview forum: "Scale Efficiently: Insights from Pretraining and Finetuning Transformers"
_ICLR.cc/2022/Conference — ICLR 2022 Poster_

### Official Review · Reviewer_C5AB · 2021-10-18

**Correctness:** 4
**Technical Novelty And Significance:** 3
**Empirical Novelty And Significance:** 3
**Recommendation:** 6
**Confidence:** 4

**Main Review:**

Strengths
* This paper conducts solid experiments across many NLP benchmarks to demonstrate the discrependcy between upstream perplexity and downstream performance.
* This paper also shows scaling strategies differ at different compute regions, i.e., applying the same strategies at different compute regions (small vs large) has a different effect on model quality.
*  This paper presents a DeepNarrow strategy that is applicable to all model sizes and can reduce model parameters and improve speed while preserving downstream performance.

Weakness
* The discrepancy between upstream perplexity and downstream performance is somewhat related to model selection with dev set in natural language generation task. Model selection with dev set according to perplexity may underperform model selection strategy according to downstream metric, e.g. BLEU score. Therefore, the discrepancy found in this paper is not "surprising"!
* DeepNarrow strategy seems also to be discussed before in other neural architectures. In Figure 6.7 of [1], discussed Conv nets with the same parameters but different layers and found that deeper models perform better. Therefore, the novelty of this strategy is not significant.

[1] Goodfellow et al. Deep Learning. 2016

**Summary Of The Paper:**

This paper presents solid experiments across many NLP benchmarks to show that the perplexity of the upstream language model can be a deceiving indicator of downstream quality. They also show that popular language model like T5-base and T5-large are relatively inefficient and scaling strategies differ at different compute regions, i.e., applying the same strategies at different compute regions has a different effect on model quality. Finally, they give a simple DeepNarrow strategy that can be applied to different model sizes and make them efficient while preserving their performance.



**Summary Of The Review:**

This paper conducts solid experiments to support its claims, e.g. discrepancy between upstream perplexity and downstream performance and scaling strategies difference at different compute regions and DeepNarrow strategy to improve model speed and reduce parameters. Some of the findings provided in this paper is kind of intuitive and the DeepNarrow strategy has been shown in previous literature, which hurt its novelty.

---

> ### Author Response · Authors · 2021-11-20
> **Response**
>
> Thanks for the review and insightful comments!
>
> Thanks for drawing the connection to the issue of “model selection with dev set”. While this may seem not-surprising in retrospect, many NLP papers strongly based their experiments on language modeling experiments with a strong assumption that good perplexity == a good model. Here, we’re also exploring another dimension of “pretraining transfer”, since perplexity is often used to imply a higher quality LM which is often implied as a better starting point for transfer learning. We believe that explicitly calling this out is an important contribution.
>
> We agree that depth vs width is a long standing problem in deep learning research and there has been work that studied this within the scope of other architectures. However, we feel that the more interesting point here (in our opinion) is that most of the other knobs do not influence the pareto-frontier as much as depth and how different this might be at different compute regions. We also added discussions around Levine et al.’s Interplay of Depth and Width in self-attention in the revision (section 4.5)
>
> Once again, thanks for the review and insightful comments!

---

### Official Review · Reviewer_8qst · 2021-10-21

**Correctness:** 3
**Technical Novelty And Significance:** 2
**Empirical Novelty And Significance:** 3
**Recommendation:** 6
**Confidence:** 3

**Main Review:**

Strengths: This paper is the first one to discover the scaling insights on both pretraining and fine-tuning Transformers. The authors perform comprehensive experiments and provide insights from different perspectives. Even though some insights are not surprising, this paper provides evidence to support some previous intuitions.

Weaknesses: Since this paper focuses on empirical conclusions, the novelty is somewhat weak. Furthermore, the experiment results may be effected by the randomness and I don't find the error bar in the paper, which is the most concern of mine. Also, I have the following questions about the paper.

1. In the caption of Figure 1, I don't see variable P (default model parallelism) in the table. Can you provide some explanation about it?
2. As discussed, do you perform repeated experiments to avoid the influence of randomness? I guess the main conclusions in your paper like Figure 1 should be performed repeatedly.
3. T5 is a sequen-to-sequence model. I am curious about your conclusions for different models like ViT or BERT. Can you perform some experiments on different models to verify the generalization of your conclusions?

**Summary Of The Paper:**

This paper presents scaling insights from pretraining and fine-tuning Transformers empirically. The main findings are as follows: (i) model shape matters for downstream tasks; (ii) scaling protocols operate differently at different compute regions; (iii) T5-base and T5-large are not Pareto-efficient. They will publicly release over 100 pretrained checkpoints for further research.

**Summary Of The Review:**

The paper study the empirical findings of scaling insights for Transformers on both pertaining and fine-tuning and provide a list of conclusions. Even though the conclusions are not surprising to me, I think this paper provides some evidence to prior works and future works. One main concern of mine is the randomness of the paper since this paper performs comprehensive experiments and concludes insights based on the results.

---

> ### Author Response · Authors · 2021-11-20
> **Response**
>
> Thanks for the review and insightful feedback.
>
> Regarding your questions and comments:
>
> - Regarding P, thanks for raising this! We have fixed this in the revision (added P to the table).
> - Our experiments follow the T5 paper’s setup, and variance across multiple runs is reported in (Raffel et al. 2020). Generally, this is quite low for superglue and glue (0.2 to 0.4) which is pretty small compared to the scale of results (73-80) for Figure 1. Moreover, In Narang et al, (which has a similar setup as ours), (https://arxiv.org/pdf/2102.11972.pdf), the standard deviation of upstream perplexity is also extremely low (+-0.005). Given that we use the same open source T5 code to run our experiments, along with the same pretraining and downstream corpus, we believe this is sufficient evidence that our results are also robust to noise and are not high variance.
> - Regarding generalizability to other architectures, we presented results on ViT in section 4.5. While an entire repeat of all experiments in the vision domain is probably out of scope for this paper, but we did show that the main results do transfer. Regarding BERT, we prioritized ViT as this was a more diverse comparison as compared to T5 given that it belongs to another domain. Recent work has also shown that T5 encoder produces similar results to BERT (https://arxiv.org/abs/2110.08426). Hence, we prioritized getting ViT results over BERT.
>
> Once again, thanks for the review!

---

> > ### Comment · Reviewer_8qst · 2021-11-29
> > **Thank you for the response**
> >
> > I have read all comments and the author's response. My main concerns have been addressed and I keep my original score.

---

### Official Review · Reviewer_SEDW · 2021-11-02

**Correctness:** 3
**Technical Novelty And Significance:** 2
**Empirical Novelty And Significance:** 3
**Recommendation:** 5
**Confidence:** 4

**Main Review:**

**Strengths**

* The paper is clearly written and the figures and tables are illustrative.
* The empirical efforts on extensive experiments deserve to be appreciated. The model checkpoints and codes will be publicly released to the community.

**Weaknesses**

* The novelty of this paper is limited.
  * Regarding the scaling laws: the motivation of this paper is that [1] only consider the pre-training performance (loss/perplexity on upstream datasets) and does not investigate the downstream performance. The authors try to fix the bias of the previously established scaling law by investigating the downstream performance. Incremental experiments investigating the downstream performance are provided. Besides, the authors do not provide further quantitative analysis like that in [1] under this new setting.
  * Regarding the proposed DeepNarrow strategy: based on the observation of the experiments, the authors propose to preferentially increase the model's depth when scaling the models. However, this finding has been discussed in [2] and the authors of [2] provide both theoretical and empirical analysis on the depth-width tradeoff of Transformer models. The authors should include this work and discuss it.

[1] Kaplan et al. "Scaling laws for neural language models." arXiv preprint arXiv:2001.08361 (2020).

[2] Levine et al. "The Depth-to-Width Interplay in Self-Attention." NeurIPS 2020.


**Summary Of The Paper:**

This paper aims at providing insights from scaling Transformers for pre-training and finetuning. Based on extensive experiments involving pre-training and fine-tuning over different transformer configurations, the authors find that the model shape matters a lot besides model size when considering the downstream performance, and scaling strategies differ at different compute regions. With these new findings, the authors propose the DeepNarrow scaling strategy and verify it on additional experiments including tasks in different domains (language and vision).

**Summary Of The Review:**

Considering the novelty of this paper, I vote for 'weak rejection'. If the authors can addressed my concerns, I would like to increase my score.

---

> ### Author Response · Authors · 2021-11-20
> **Response**
>
> Thanks for the review and feedback, along with the time spent on reviewing our work. We greatly appreciate it.
>
> To address the comment from the reviewer on using techniques from [1] in the setup we have, we added a section to the updated paper (Appendix A), where we fit  power law curves to model the trend of scaling for pre-training and fine-tuning performance versus parameter, given the models we used in Figure 1 of our paper. Note that [1] applies such a technique on other factors than parameter size, like data size or training step, which is not applicable here, given we fix these factors in our experiments and change other knobs.
>
> In our setup, we cannot directly compare the values of the function we fit for upstream and downstream, since the performance metrics are completely different (perplexity vs accuracy). However, we observed a much higher fitting error when modeling the fine-tuning performance compared to modeling the pre-training performance. This once again, confirms that unlike pre-training performance, fine-tuning performance is not easily predictable based on parameter size and the shape of models plays an important role on how it performs on the target task.
>
> We have also added discussions about the paper “The Depth-to-width interplay in self-attention” [2] in section 4.4. to the paper, Specifically, we highlight the places where this work presents novel insights over this work. While Levine et al. share similar prescriptions as our work, there are differences. Firstly, empirical analysis seem to be only limited to 100M parameters in Levine et al. (Figure 3). The authors seem to only rely on extrapolation to derive optimal fit to larger models. On the other hand, our work actually provides data points at 10B+ models. Moreover, we also find that scaling protocols may slightly differ at different compute regions and they might not be easily extrapolated. In our revised paper, we also discuss how our findings differ and where their extrapolation may not align with our empirical results.
>
> Finally, the results in Levine et al. are only restricted to language modeling experiments (similar to Kaplan et al.) with an artificially small vocab (2K). On the other hand, the results presented in this work reflect practical usage of these large scale models at both upstream and transfer (downstream) setups. Overall, we believe that our work has sufficient merit (and novel insights) that are beneficial to the community.
>
> Thanks for the review!
>
> [1] Kaplan et al. "Scaling laws for neural language models." arXiv preprint arXiv:2001.08361 (2020).
>
> 2] Levine et al. "The Depth-to-Width Interplay in Self-Attention." NeurIPS 2020.

---

### Official Review · Reviewer_826H · 2021-11-02

**Correctness:** 4
**Technical Novelty And Significance:** 1
**Empirical Novelty And Significance:** 3
**Recommendation:** 8
**Confidence:** 4

**Main Review:**

Strengths:
- The findings in the paper are useful for researchers in this domain from two folds: Firstly, by open-sourcing the checkpoints, others can take advantage of the trained models to foster research in various NLP applications. Secondly, the heuristics provided can potentially guide the design of better architectures for downstream tasks.

Weaknesses:
- The plots are often hard to read and have very bad visualization. As an example, the labels in Figure2-c are unintelligible. As such, it is hard to drive conclusions from the plots, rather the reader has to refer only to the text, which is not adequate. It would be great if the authors improve the plots and ensure that all text/labels can be easily read.

**Summary Of The Paper:**

This paper provides various ablation studies to show the effect of scaling various architecture parameters on the performance of a downstream task. The studies are performed on encoder-decoder transformer architectures that follow the T5 architecture as the backend. As a result of the ablation studies, the authors provide a heuristic on how to design architectures that lie on a better Pareto front compared to the previously proposed T5 architectures.

**Summary Of The Review:**

Please see above.

---

> ### Author Response · Authors · 2021-11-20
> **Response**
>
> Thanks for the insightful feedback and comments, along with taking the time to review our paper! We are glad you liked our paper.
>
> Regarding the plot quality, the plot quality in the current paper is already in high resolution. We feel that they might be difficult to read as we had to make them smaller so that we could fit the commentary and text in the paper. For example, the issue pointed out in Figure 2c is not a resolution issue but a plot size issue which makes the labels cramped.
>
> That said, we do acknowledge and understand where you are coming from and would propose to also release extremely high resolution PDFs that are fully expanded (i.e., 1 plot per page)  in our code release so that users and readers could always reference the super high quality plots if they need to. We are also open to other suggestions!
>
> Once again, thanks for the review!

---

### Author Response · Authors · 2021-11-20
**General Response and Summary of updates**

Thanks to all reviewers for the hard work and effort spent reviewing our manuscript. We are greatly appreciative of the insightful comments and feedback.

We have updated the manuscript with the following changes after taking into consideration the feedback provided.

- We added additional quantitative analysis (scaling law plots) in the appendix similar to Kaplan et al. (as requested by Reviewer SEDW)
- We added discussions on Levine et al’s Depth-to-Width interplay to the main text. (as requested by Reviewer SEDW).
- We moved “diverse NLP experiments” to the Appendix to make room for additional discussions on Depth-width interplays.
- We fixed a comment about parallelism missing from the table (requested by Reviewer 8qst)

Once again, thanks to all reviewers for their comments!

---

### Decision · Program_Chairs · 2022-01-20

**Decision:**

Accept (Poster)

**Comment:**

The results reported in this paper and the model checkpoints released are of interest and broad utility to the community in the opinion of the NLP.  While one reviewer was somewhat negative, most reviewers were in favor of acceptance of this paper, which expands the results from [1] to downstream tasks. The AC therefore recommends acceptance.